# Appropriateness of Prescribing Transmucosal Immediate-Release Fentanyl in the Emergency Room, During Hospitalization, and at Discharge: A Retrospective Study

**DOI:** 10.3390/ph17121609

**Published:** 2024-11-28

**Authors:** Gary Punjabi, Elena Ramírez

**Affiliations:** Clinical Pharmacology Department, La Paz University Hospital-IdiPAZ, School of Medicine, Autonomous University of Madrid, 28029 Madrid, Spain; gary.punjabi@estudiante.uam.es

**Keywords:** transmucosal immediate-release fentanyl, appropriateness, adequacy, indication, emergency room, discharge, off-label prescribing, oncological, hospitalization

## Abstract

**Background/Objectives**: This study evaluated the appropriateness of transmucosal immediate-release fentanyl (TIRF) prescriptions in a Madrid emergency room during 2019 and 2022, following a 2018 warning about off-label use. **Methods**: TIRF prescription in the emergency room search yielded 993 patients in 2019 and 1499 in 2022, of which 140 were randomized for the study, 70 in 2019, and 70 in 2022. Dose appropriateness and indication for TIRF were analyzed according to established criteria. **Results**: Despite a high prevalence of cancer diagnoses (77.9%, 109/140), only 32.9% (46/140) of patients met the appropriateness criteria pre-hospitalization. This improved to 42.5% (51/120) at discharge, but the change was not statistically significant overall. However, focusing on surviving patients reveals a significant improvement in appropriateness, increasing from 30.83% (37/120) to 42.50% (*p* = 0.002). This improvement was particularly pronounced in 2022 (*p* = 0.0269), but not in 2019 (*p* = 0.0771). Interestingly, appropriateness in patients with prior TIRF prescriptions remained relatively stable from pre-hospitalization (46.75%) to discharge (48.78%). A concerningly high proportion of patients with cancer diagnoses (68.75%) received low-dose opioid therapy (<60 MME) at discharge, and 36.8% of patients over 80 years old were co-prescribed benzodiazepines, contradicting prescribing guidelines. **Conclusions**: This study found inappropriate TIRF prescriptions were common in an emergency room setting, often due to low pre-hospital opioid doses. While hospitalization improved TIRF appropriateness in survivors, especially in 2022, concerning prescribing practices persisted. This emphasizes the need for better education and interventions to ensure safe and effective TIRF use.

## 1. Introduction

In 2018, the Spanish Agency of Medicines and Medical Devices (Spanish acronym, AEMPS) published an informative note for health professionals urging them to “Respect the conditions of authorization of the transmucosal immediate-release forms of fentanyl (TIRF), whose authorized indication is breakthrough pain of oncological origin treated with baseline opioid analgesic. Assess the need for treatment and the use of other therapeutic alternatives in patients already on treatment with transmucosal immediate-release fentanyl for non-cancer pain. In these patients, their potential for abuse can be assessed according to the questionnaires available for this purpose and they should be adequately informed of the risk of abuse and dependence associated with its use” [1].

Despite this warning, inappropriate TIRF prescribing continues to be a concern. A study in the Valencia region of Spain evaluated the impact of different interventions (two medication reviews and a safety warning letter) on off-label TIRF prescriptions for non-cancer pain and breakthrough cancer pain in patients without chronic pain therapy. The study found that while the interventions initially reduced inappropriate prescribing, these effects were modest and temporary [2].

This issue is not unique to Spain. In 2022, 13 high-income countries accounted for 78.2% of the global fentanyl consumption. Germany (19%), Spain (12.9%), and the United States (12.8%) were the top consumers, with France (6.4%) and the Netherlands (4.2%) also showing significant use [3].

High consumption is coupled with alarming rates of off-label TIRF prescriptions. In France (2019), 51.8% TIRF prescriptions were off-label, with the majority (81.7%) being for patients without a cancer diagnosis [4]. Similarly, in the Netherlands (2019), 74.9% of the TIRF prescriptions in primary care were off-label, with a substantial proportion (65.2%) not receiving maintenance opioid therapy [5].

In Spain however a study of the incidence of TIRF between 2012 and 2017 found that 27% had a prescription unrelated to cancer [6]. Another study at the 12 de Octubre Hospital in 2017 found that 31.8% TIRF hospital use were non-cancer diagnosis [7].

Growing evidence highlights the concerning trend of off-label TIRF use and its potential consequences. Analysis of EudraVigilance data (1992–2018) revealed that the majority (81.9%) of reported cases of suspected fentanyl-associated abuse, dependence, or withdrawal occurred in patients without a cancer diagnosis, suggesting widespread off-label use [8]. These findings are particularly alarming given the broader context of increasing opioid prescriptions and their associated risks. The rise in opioid prescriptions has been linked to a higher number of opioid-related deaths [9], and opioid prescriptions at discharge are known to contribute to chronic opioid use [10].

Therefore, this study investigates a critical question: does hospitalization, with its potential for medical intervention and reassessment, lead to modifications in prescribing practices and improved appropriateness of TIRF treatment for patients initially receiving it in the emergency room? This analysis includes comparing appropriateness in 2019 versus 2022. Secondly, the study examined TIRF utilization patterns throughout the hospitalization process (before arriving at the emergency room, emergency room, inpatient stay, and discharge), focusing on dosage appropriateness and how prescribing practices vary across different patient demographics (age, sex, diagnosis, etc.). By examining this question, we aim to contribute to a better understanding of TIRF utilization and identify potential areas for intervention to promote safer and more appropriate prescribing.

## 2. Results

A search for TIRF prescription in the emergency room yielded 993 patients in 2019 and 1499 in 2022, of which 140 were randomized for the study, 70 in 2019, and 70 in 2022. Figure 1 shows the study flowchart.

### 2.1. Appropriateness of TIRF Treatment in the Hospital

This study included 140 patients, of whom 77.9% (109/140) had a cancer diagnosis. Prior to hospitalization, 81.4% (114/140) were receiving TIRF, and 40.7% (57/140) had a baseline (pre-hospitalization) opioid dose exceeding 60 MME. Based on these factors, 32.9% (46/140) met the criteria for appropriate TIRF use. Sadly, 20 patients (14.2%) died during hospitalization. Among the surviving patients, TIRF prescriptions at discharge increased to 60.83% (73/120). Of these, 69.9% (51/73) met the appropriateness criteria, representing 42.5% (51/120) of all surviving patients. This change in appropriateness was not statistically significant (*p* = 0.109). Among surviving patients, appropriateness to TIRF indication significantly improved from 30.83% (37/120) pre-hospitalization to 42.50% (51/120) at discharge (*p* = 0.002).

The evolution of TIRF MME across the different phases, excluding those who died, using a paired samples *t*-test showed that the MME in the emergency room 6.42 (5.56) was higher than the others (*p* < 0.0001): pre-hospitalization MME: 3.50 (SD 5.03), *p* < 0.0001; MME during hospitalization: 4.39 (SD 5.70), *p* = 0.02; and MME at discharge: 4.07 (SD 4.93), statistical significance (*p* = 0.004) (Figure 2). The difference in total MME between pre-hospitalization and at discharge was a mean (SD) of +16.35 (SD 50.02). The difference in TIRF dose after hospital stay, i.e., the difference between TIRF MME at discharge and prior (pre-hospitalization) TIRF MME, was a mean (SD) of −0.195 (SD 4.71). Excluding the deceased, the average was +0.56 (SD 4.21).

The correlation analysis showed a high correlation (0.87) between total pre-hospitalization overall MME (baseline + TIRF MME) and at discharge, and 0.64 for pre-hospitalization TIRF MME and at discharge. Age was negatively correlated with the time of prescription by −0.21 (Figure 3).

Cancer diagnosis was associated with shorter TIRF prescription duration. Higher TIRF MME and DDD was seen for both pre-hospitalization and discharge (*p* = 0.001). It was also related to lower frequency of mental health history (*p* = 0.002).

Patients with appropriateness to TIRF indication had a higher pre-hospitalization TIRF MME and DDD (*p* < 0.001), a higher overall MME for both pre-hospitalization and at discharge (*p* <0.001), and an increased concomitant use of beta-blockers (*p* = 0.025). Of the 20 TIRF prescriptions at discharge considering the prior baseline treatment, only one patient met the indication criteria, but if the baseline treatment at discharge is considered, this number increases to 5 (25%). The analysis of TIRF dose changes in the 16 patients with improved appropriateness showed a variety of adjustments, including dose increases (N = 2), deprescribing (N = 3), new prescriptions (N = 4), dose reductions (N = 1), and no change (N = 6).

Of the 140 patients in the study, 58.57% (82/140) were receiving TIRF prior to their emergency room visit. Among these patients, 72 had an oncological diagnosis, and 48.78% (40/82) met the appropriateness criteria for TIRF use. At discharge, the proportion of patients receiving TIRF increased to 60.83% (73/120). Of these, 61 had an oncological diagnosis, but only 46.75% (34/73) met the appropriateness criteria.

### 2.2. Comparison of 2019 vs. 2022

An overview of patient demographics, hospitalization information, fentanyl prescription patterns, and comedication use for the years 2019 and 2022 is presented in Table 1. A significant difference in pre-hospitalization TIRF MME was observed between the two years, with a higher average in 2019 (4.60 MME) compared to 2022 (2.76 MME) (*p* = 0.030).

While overall TIRF use (MME) at discharge, excluding the deceased, did not significantly differ from pre-hospitalization levels, the analysis of TIRF doses specifically revealed significant changes between the two periods (*p* = 0.048). Interestingly, TIRF MME decreased by an average of 0.23 (SD 3.79) in 2019, while it increased by an average of 1.28 (SD 4.47) in 2022.

From 2019 to 2022, the proportion of patients receiving low-dose opioids (0-60 MME) increased (24.3% vs. 48.6%, *p* = 0.005). The proportion of patients with concomitant antidepressant use increased from 11.4% in 2019 to 32.9% in 2022 (*p* = 0.004).

There were also changes in the distribution of patients by age range, with an increase in the proportion of patients aged 65–79 years (24.3% vs. 54.3%) and a decrease in those over 80 years of age (37.1% vs. 17.1%) (*p* = 0.003). However, the mean age remained similar (69.43 vs. 69.63 years).

The analysis of appropriateness for TIRF indication revealed interesting trends between 2019 and 2022. In 2019, appropriateness increased slightly from 35.7% (25/70) pre-hospitalization to 42.1% (24/57) at discharge, but this change was not statistically significant (*p* = 0.462). A similar trend was observed in 2022, with appropriateness rising from 30% (21/70) to 42.8% (27/63) (*p* = 0.291). Focusing on surviving patients and using paired samples analysis, we found that in 2019, appropriateness increased from 26.32% (18/57) pre-hospitalization to 42.10% (24/57) at discharge, but again, this difference was not statistically significant (*p* = 0.0771). However, in 2022, a significant improvement was observed among survivors, with appropriateness rising from 30.16% (19/63) pre-hospitalization to 42.86% (27/63) at discharge (*p* = 0.0269) (Table 2).

### 2.3. Patient Characteristics

The characteristics of the patients by age are shown in Appendix A. Prescription duration varied significantly across age groups (*p* = 0.036). Patients aged 45-64 received longer prescriptions (mean 302.56 days, SD 692.85) compared to other age groups (*p* = 0.036).

Smokers were significantly younger, with a mean age of 61.81 (SD 11.40) compared to 72.84 (SD 13.68) for non-smokers (*p* < 0.001). Patients with mental health conditions had more episodes (*p* < 0.001), longer prescription (*p* < 0.001), lower mean age (*p* = 0.006), lower frequency of cancer diagnosis (*p* = 0.002), higher use of antidepressants (*p* = 0.002), and higher overall MME both pre-hospitalization and at discharge (*p* < 0.001 and *p* = 0.010, respectively). Benzodiazepine use was associated with a higher pre-hospitalization TIRF MME and DDD (*p* = 0.048) and higher use of antidepressants (*p* = 0.011). Notably, 36.8% of patients over 80 years old used benzodiazepines. Antidepressant use was linked to an increased use in 2022 (*p* = 0.004), higher overall MME (*p* < 0.001), and greater frequency of >60 MME baseline opioid treatment (*p* = 0.043). Gabapentinoid use correlated with both pre-hospitalization overall MME (*p* = 0.009) and overall MME at discharge (*p* = 0.001). All cases of opioid intolerances occurred in female patients (*p* = 0.012). Patients with opioid intolerance had a lower mean age of 68.82 (SD 13.87) compared to 77.82 (SD 12.63) for those without (*p* = 0.040).

### 2.4. Characteristics of TIRF Adverse Effects

Of the 7 patients with adverse effects described in their medical history, 5 had an oncological diagnosis, where 3 of these were appropriate to the indication. Of the other two, one was adjusted to the indication at discharge and another, without prior treatment, was prescribed opioids at low doses.

The mild or moderate symptoms are as follows:“Constipation” (3 patients): (1) no change at discharge (transdermal fentanyl dependence) (appropriate to indication).“MME increase, 23.9” (2 patients): 3.9 TIRF MME (not appropriate).“MME increase, 53.9” (3 patients): 3.9 TIRF MME (later adjusted to indication).“Drowsiness”: death (appropriate).

The severe symptoms are as follows:“Constipation, asthenia and emetic syndrome”: increase of 52.8 MME (not appropriate: no oncological diagnosis).“Acute hyperactive confusional syndrome”: reduction of 30 MME (not appropriate: no oncological diagnosis).“Respiratory depression in probable relation to erratic absorption of opioids”: reduction of 30 MME (appropriate).

Notably, no patients were found to be using CYP3A4 inducers, nor were any opioid abuse risk assessments (using tools like SOAPP-R or ORT) documented. Only three patients had a documented history of opioid dependence.

### 2.5. Characteristics of Patients Without Prior TIRF

The characteristics of the patients without pre-hospitalization TIRF are shown in Appendix A. They had lower overall MME for both pre-hospitalization (*p* < 0.001) and at discharge (*p* = 0.007). A smaller proportion had a cancer diagnosis (63.8% vs. 87.8%, *p* = 0.002). Fewer met the criteria for appropriate TIRF use (10.3% vs. 48.8%, *p* < 0.001). They were more likely to have a single episode (98.3% vs. 76.8%, *p* = 0.005). They had a lower rate of antidepressant use (10.3% vs. 30.5%, *p* = 0.009).

### 2.6. Patient Characteristics by Hospitalization Status

Of the 140 patients in this study, 47 were not hospitalized in a ward. These non-hospitalized patients differed from those who were hospitalized in the following several key ways: They had a significantly lower mean pre-hospitalization TIRF MME and DDD (*p* = 0.046). They were also less likely to be receiving high-dose baseline opioid treatment (>60 MME) (*p* = 0.049). They had a significantly higher proportion of TIRF prescriptions at discharge (31.8% vs. 7.9%, *p* = 0.010). Further details on characteristics by hospitalization status are provided in Appendix A.

### 2.7. Characteristics of Prescriptions, Deprescriptions, and Deaths

Patients receiving TIRF prescriptions at discharge had a significantly lower pre-hospitalization overall MME compared to those who did not receive TIRF (median of 12.5 vs. 42.11, *p* < 0.001). Interestingly, only 5 out of 20 (25%) of these new TIRF prescriptions met the appropriateness criteria when considering baseline MME at discharge, and only 1 patient met the criteria based on pre-hospitalization MME.

Despite TIRF deprescription, overall MME at discharge increased by a mean of 30.57 MME. In the non-deprescription group, the mean overall MME at discharge was 14.91. In the deprescription group, 7/11 patients met the indication at discharge, while only 4 did when considering pre-hospitalization MME. Overall, 100% of patients had an oncological diagnosis, suggesting a possible substitution of TIRF with longer-acting opioids.

A further analysis of patient outcomes is detailed in Appendix A, which provides characteristics of prescriptions, deprescriptions, and deaths.

Interestingly, deceased patients did not significantly differ from the rest of the cohort, except for a higher dose of TIRF during hospitalization. However, this difference disappeared when excluding patients without ward hospitalization, likely due to the lower proportion of non-hospitalized patients in the deceased group (5% vs. 36.7%, *p* = 0.04) [3].

## 3. Discussion

### 3.1. Evolution of Appropriateness During Hospital Visits

We analyzed the percentage of patients meeting the criteria for appropriate TIRF use (breakthrough oncological pain and baseline opioid treatment exceeding 60 MME) to assess if appropriateness improved throughout hospitalization. Despite 80% of patients having an oncological diagnosis, 68.75% did not meet the >60 MME requirement at discharge. Furthermore, of the 20 TIRF prescriptions at discharge, only 5 met criteria, and 4 were non-cancer pain.

This unexpectedly high number of patients on low-dose opioids (<60 MME) may be attributed to the gradual initiation of opioid treatment, particularly with transdermal fentanyl. Transdermal fentanyl is often initiated at a low dose (12 mcg/h, equivalent to 30 MME) and titrated upward to 25 mcg/h (60 MME) as needed. However, this practice raises concerns regarding opioid tolerance. If the stricter FDA TIRF Risk Evaluation and Mitigation Strategy were followed, which requires prescribers to document opioid tolerance for every outpatient TIRF prescription, 75% of these observed prescriptions would be considered invalid. This discrepancy highlights a critical issue: clinicians are prescribing TIRF to patients who may not be opioid-tolerant, putting them at risk of serious complications, including respiratory failure and death [11]. Additionally, patients may experience more frequent or severe breakthrough pain episodes, requiring additional analgesia and potentially impacting their daily activities and overall wellbeing. These patients may require closer follow-up with their primary care physicians to optimize pain management, adjust treatment plans as needed, and mitigate potential risks. Telemedicine could play a valuable role in facilitating this ongoing monitoring and support. Surveying medical professionals about their TIRF prescribing patterns could provide valuable insights into the factors driving these practices and inform interventions to improve patient safety.

While the proportion of patients meeting the appropriateness criteria for TIRF use increased from 32.9% pre-hospitalization to 42.5% at discharge, it is crucial to note that the initial appropriateness was strikingly low. Even when focusing only on surviving patients, the improvement from 30.83% to 42.50% at discharge highlights a significant opportunity to optimize TIRF prescribing practices in this population.

This study found a significantly higher TIRF dose administered in emergency rooms compared to during hospitalization. This discrepancy may be attributed to a potential initial lack of knowledge regarding appropriate TIRF dosing. Dosage adjustments often occur based on the number of rescue doses required, highlighting the need for improved initial dose optimization.

In our study, 87.8% of patients who already had prior TIRF had an oncological diagnosis. This figure represents an improvement compared to a 2017 in-hospital prescription study, where 68.2% of prescriptions were for cancer patients [7]. In another study also from 2017, in primary care, 73% had an oncological diagnosis [6]. However, it is important to consider these results in the context of our study’s Berkson or admission bias.

The average pre-hospitalization TIRF MME was higher in 2019 than in 2022. While overall opioid use (MME) decreased at discharge, TIRF MME actually increased, alongside a higher number of TIRF prescriptions. This trend did not reach statistical significance. One possible explanation is that the higher initial TIRF MME in 2019 led to better appropriateness, allowing for greater dose reduction at discharge. Specifically, TIRF MME decreased by an average of 0.23 in 2019, while it increased by an average of 1.28 in 2022 (*p* = 0.048). Despite this dose increase in 2022, appropriateness did not worsen, as the percentage of patients meeting the indication criteria also increased (*p* = 0.0269).

Concerningly, 36.8% of patients over 80 years old were prescribed benzodiazepines, contradicting recommendations to avoid co-prescribing these medications with opioids in elderly populations [12]. This practice raises concerns about potential adverse events and drug interactions.

The use of antidepressants, as well as gabapentinoids, was associated with higher overall MME, which is consistent with previous studies [13]. This highlights the complex interplay between pain management, mental health, and opioid use.

A history of mental health was associated with a longer duration of opioid treatment, as previously described [14]. This underscores the need for careful monitoring and individualized pain management strategies in patients with mental health conditions.

Despite recommendations for opioid risk assessment, a revision of primary care and hospital records revealed that no patient in our study underwent an opioid abuse risk questionnaire. This omission may be due to a lack of awareness of these questionnaires or a perception among clinicians that they are not useful. However, it is important to emphasize that while these standardized tools may have limitations in specificity and sensitivity, they can still provide valuable guidance in identifying patients at risk [12].

### 3.2. Study Limitations

The study analyzed prescriptions and not dispensations, which may lead to a possible overestimation. On the other hand, the study only recorded prescriptions from the National Health System, which could lead to underestimation by not including private prescriptions. This applies to both opioid treatment and other concomitant treatments.

The methodology was based on the review of medical records, which can generate human errors. The effect on pain management or quality of life was not evaluated, nor were genetic or pharmacokinetic factors considered that may influence the rate of adverse effects. There may be a significant underreporting of adverse effects since they are not always recorded in the medical record, especially those of a milder nature. The data used in this study are limited to a single center, a large, complex university hospital, and therefore the study’s conclusions may not be generalizable to other centers. To further validate these results and gain a more comprehensive understanding of TIRF prescribing practices, future research should include prospective, multi-center studies with validated pain and quality of life measures, as well as qualitative studies exploring clinician perspectives and experiences.

Only TIRF, a subgroup of prescribed opioids, was analyzed however they have the clearest indication criteria. The use of drugs of abuse and alcohol, which considerably increases the risk of abuse or addiction, was not considered since it is not always recorded in the medical history. The conversion to MME of transmucosal fentanyl is not fully established, but it does not affect the relative comparison of doses.

### 3.3. Future Proposal

Less than half of the patients prescribed TIRF in the emergency room met the indication criteria. The reasons for this could be due to “as needed” prescriptions that are not used, the need for rapid pain relief due to the delay in care in the ER, or for other reasons that are unknown. Oncological diagnosis does not guarantee good treatment appropriateness, and its absence is usually the focus of other studies. Therefore, more research is needed on the appropriateness of opioids in cancer patients [15]. Some authors consider that the AEMPS notification on TIRF is insufficient and that other more widely used opioids should be included [16]. Further research should also be conducted on the appropriateness of transdermal fentanyl, as the guideline recommends its use only in patients who cannot take fentanyl orally [12]. In 2022, transdermal fentanyl accounted for 89.92% of fentanyl used, being the third most used opioid after tramadol with paracetamol and tramadol [17].

To improve appropriateness, the recording of off-label use of medicines should be improved [18]. Educational programs could also be considered as there is evidence that they can produce a temporary decrease in the general prescription of TIRF [19]. Better monitoring and documentation of off-label TIRF prescriptions to identify trends, assess potential risks, and inform interventions is needed. This could involve implementing electronic health record systems that flag off-label use or require justification for such prescriptions. Other studies show that the amount of opioids prescribed can be reduced simply and inexpensively by dispensing predetermined quantities through electronic medical records [20,21,22]. This could help prevent overprescribing and reduce the risk of misuse or diversion. Finally, patients requiring frequent TIRF rescues should be encouraged to consult their primary care physician, utilizing telemedicine when feasible, to ensure appropriate pain management and address potential concerns about opioid tolerance [23].

## 4. Materials and Methods

### 4.1. Setting

This retrospective observational drug utilization study was conducted at La Paz University Hospital, a tertiary care teaching hospital in Madrid, Spain. The study was approved by the hospital’s Institutional Review Board (protocol PI-5960). Due to the retrospective nature of the study, informed consent was not required.

The study included all patients who received prescriptions for TIRF (N02AB03) in the emergency room at La Paz University Hospital between 1 January 2019 and 31 December 2019, and between 1 January 2022 and 31 December 2022.

Patients were excluded if they:Were transferred to another hospital after their emergency room visit;Were treated in the day hospital or outpatient setting and not formally admitted;Received fentanyl as part of an anesthetic or sedation protocol.

### 4.2. Detection Sequence

To identify potential study participants, a search was conducted for all patients who received a TIRF prescription in the La Paz University Hospital’s emergency room during the study period. Each prescription represented a unique episode, and each episode corresponded to a single patient. In cases where a patient had multiple episodes, these were recorded.

From this initial pool, patients were randomly selected for inclusion in the study using an Excel-generated randomization sequence. Each selected patient was assigned a unique study number (documented in the Case Report Form, CRF) to maintain anonymity. Researchers then accessed and collected relevant study variables from both primary care (AP Madrid) and hospital (HCIS) electronic health records.

Finally, the predefined inclusion and exclusion criteria were applied to the randomized patient pool. For all patients meeting the inclusion criteria, study variables were systematically extracted and recorded in a dedicated CRF.

### 4.3. Study Medications and Dosage Calculations

This study focused on the use of TIRF, with the ATC code N02AB03. TIRF is available in various formulations for oral and nasal delivery.

Defined daily doses (DDDs) and morphine milligram equivalents (MMEs) were calculated to standardize medication dosages.

DDD: The World Health Organization’s (WHO) defined daily dose for sublingual fentanyl (ATC code N02AB03) is 0.6mg, which is based on adult usage for pain management [24].MME: MMEs were calculated using a conversion factor of 130, indicating that sublingual fentanyl is 130 times more potent than morphine [25].

For other medications used in the study, DDD values were obtained from the WHO Collaborating Centre for Drug Statistics Methodology [24], while MME values were sourced from the CDC [26].

### 4.4. Appropriateness

Appropriate drug use is defined by strong evidence supporting its use in a given indication, good tolerability, and cost-effectiveness. Conversely, inappropriate prescribing occurs when the potential harms outweigh the benefits, particularly when safer or more effective alternatives exist [27].

To assess the appropriateness of TIRF prescriptions, the study adhered to the Spanish Consensus Guideline for the Good Use of Opioid Analgesics [12]. These guidelines advise against using TIRF for chronic non-cancer pain due to the high risk of abuse and addiction and recommend obtaining informed consent for any use outside the authorized product information. According to the product information, TIRF is indicated for adult patients with chronic cancer pain who are already receiving opioid maintenance therapy equivalent to at least 60 mg of oral morphine daily [28].

### 4.5. Evaluation Criteria

Main Variable: Appropriateness of TIRF prescription: This was determined by indication (the presence or absence of a cancer diagnosis in the last 5 years) and baseline opioid use (whether the patient was receiving opioid treatment with a baseline dose greater than 60 MME, less than 60 MME, or no baseline opioid treatment).

Secondary Variables: The following are the secondary variables: patient characteristics (sex, age, cause of pain, history of smoking in the last 5 years, history of mental health disorders and allergies, and previous diagnoses); hospitalization (date of emergency room visit, date of hospitalization, outcome (discharge or death), hospital department/ward, diagnoses); TIRF (formulation, dosage measured in DDDs and MMEs, including prescribed dose and dose regimen, start and end date, number of MMEs and DDDs at different time points (before arriving at the emergency room (pre-hospitalization), emergency room, hospitalization, discharge), indication, and dose changes); concomitant medications (benzodiazepines, Z drugs, antirheumatics, antidepressants, beta blockers, oral antidiabetics, gabapentinoids, CYP3A4 inhibitors, and CYP3A4 inducers) [29]; record of any assessment with an addiction risk scale (such as ORT and SOAPP tools); time in days since first TIRF prescription; and allergy or intolerance to NSAIDs or other opioids.

### 4.6. Expected Sample Size and Basis for Its Determination

This study aimed to recruit 140 patients, 70 from 2019 and 70 from 2022. This sample size was determined using a power analysis for a paired samples study comparing MME and DDD of fentanyl at various time points (before, during, and after hospitalization). With an alpha of 0.05, power of 80%, and estimated medium effect size of 0.5, the calculated sample size was 128 [30]. This was increased to 140 to account for potential data loss.

### 4.7. Description of the Statistical Analysis

Statistical analysis was performed using RStudio version 2024.04 (Posit team, 2024). Categorical variables were summarized using frequencies and percentages, with chi-square tests (chisq.test or prop.test functions) used for comparisons. Quantitative variables were summarized using means and standard deviations for normally distributed data, and medians with interquartile ranges for non-normally distributed data. Normality was assessed using the Lilliefors (Kolmogorov–Smirnov) test.

Age was categorized into four groups: 18–44 years, 45–64 years, 65–80 years, and >80 years.

Hypothesis testing was conducted using the following:Student’s *t*-test: For comparisons between two groups with normally distributed data (independent or paired samples).ANOVA: For comparisons between more than two groups with normally distributed data.Kruskal–Wallis test: For comparisons between two or more groups with non-normally distributed data.

Additionally, correlation analysis was performed to examine the relationships between age and transmucosal immediate-release fentanyl (TIRF) dosage (expressed as morphine milligram equivalents, or MME) at various time points (pre-hospitalization, in the emergency room, during hospitalization, and at discharge).

## 5. Conclusions

In conclusion, this study reveals a concerningly low rate of appropriate TIRF prescriptions among patients presenting to the emergency room, primarily due to inadequate baseline opioid therapy. While overall appropriateness did not significantly improve at discharge, hospitalization was associated with increased TIRF appropriateness among surviving patients, particularly in 2022. However, the percentage of patients with prior TIRF prescriptions meeting indication criteria remained largely unchanged. This underscores the need for improved adherence to TIRF prescribing guidelines and targeted interventions to optimize prescribing practices and ensure the safe and effective use of TIRF in this population.

## Figures and Tables

**Figure 1 pharmaceuticals-17-01609-f001:**
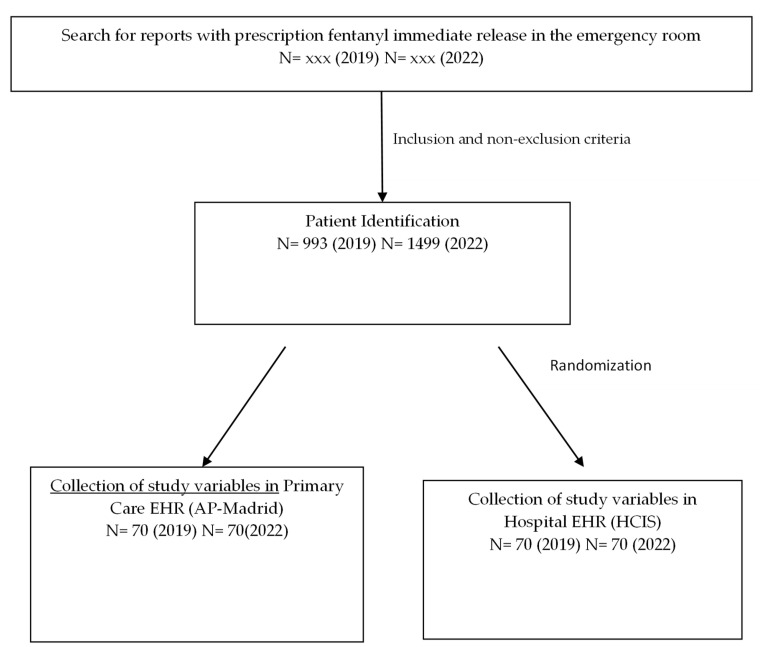
Study flowchart. Note: EHR, electronic health records.

**Figure 2 pharmaceuticals-17-01609-f002:**
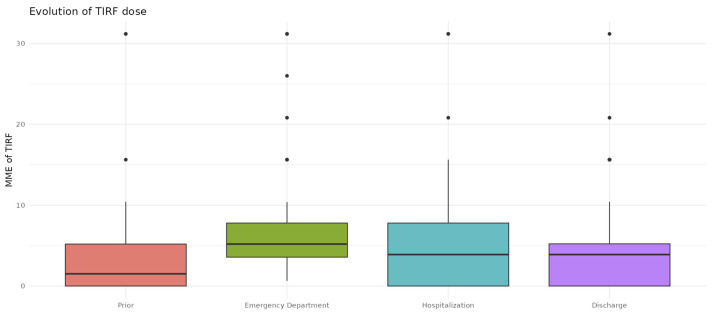
A box plot of TIRF’s MME evolution. A bar graph representing box TIRF MME contained between the interquartile range 1 and 3. The median is represented by a black line and outliers by black dots. Deceased patients are excluded from this graph.

**Figure 3 pharmaceuticals-17-01609-f003:**
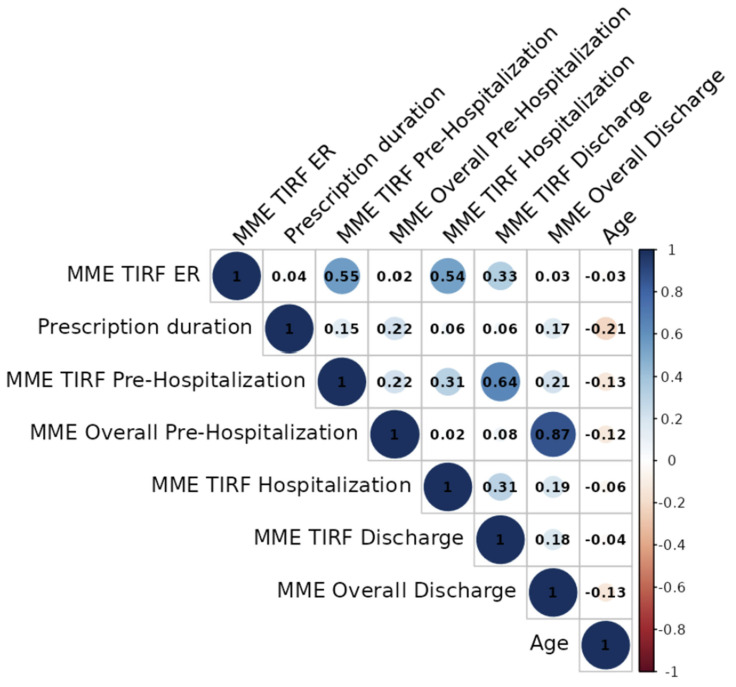
Correlation matrix. Correlation of −1 to 1 between numerical variables, excluding deceased patients. DDD not shown as it is proportional to MME.

**Table 1 pharmaceuticals-17-01609-t001:** Characteristics of patients, hospitalization, and fentanyl prescription (2019 and 2022).

Variable	All	2019	2022	*p*-Value
Number of patients	140	70	70	-
Number of episodes (%)				<0.001 ***
1	120 (85.7)	50 (71.4)	70 (100.0)	
2	12 (8.6)	12 (17.1)	0 (0.0)	
3	5 (3.6)	5 (7.1)	0 (0.0)	
5	3 (2.1)	3 (4.3)	0 (0.0)	
ER TIRF MME (mean (SD))	6.58 (5.40)	7.30 (6.02)	5.86 (4.64)	0.116
ER TIRF DDD (mean (SD))	0.84 (0.69)	0.94 (0.77)	0.75 (0.60)	0.116
Average Prescribing Time in Days (mean (SD))	141.24 (409.94)	197.29 (551.66)	85.20 (166.84)	0.106
Prior TIRF MME (mean (SD))	3.68 (5.03)	4.60 (6.29)	2.76 (3.13)	0.030 *
Prior TIRF DDD (mean (SD))	0.47 (0.65)	0.59 (0.81)	0.35 (0.40)	0.030 *
Prior Overall MME (mean (SD))	68.59 (96.23)	73.47 (107.12)	63.72 (84.45)	0.551
MME TIRF in Hospitalization (mean (SD))	4.83 (5.71)	5.37 (5.51)	4.28 (5.90)	0.263
Hospitalization TIRF DDD (mean (SD))	0.59 (0.72)	0.64 (0.69)	0.55 (0.76)	0.458
Discharge TIRF MME (mean (SD))	3.49 (4.78)	3.28 (5.24)	3.69 (4.31)	0.613
Discharge TIRF DDD (mean (SD))	0.45 (0.61)	0.42 (0.67)	0.47 (0.55)	0.613
Discharge Overall MME (mean (SD))	74.53 (87.16)	77.12 (91.28)	71.94 (83.42)	0.727
Adverse effects (%)	7 (5)	5 (7.1)	2 (2.9)	0.438
TIRF Changes (%)				0.516
Increase	10 (7.1)	4 (5.7)	6 (8.6)	
Prescription	20 (14.2)	7 (10.0)	13 (18.6)	
Reduction	7 (5)	4 (5.7)	3 (4.3)	
Deprescription	11 (7.9)	6 (8.6)	5 (7.1)	
Deaths	20 (14.2)	13 (18.6)	7 (10.0)	
No changes	72 (51.4)	36 (51.4)	36 (51.4)	
NSAID intolerance (%)	11 (7.9)	6 (8.6)	5 (7.1)	1.000
Opioid intolerance (%)	7 (5.0)	3 (4.3)	4 (5.7)	1.000
Age (mean (SD))	69.53 13.95	69.43 (15.52)	69.63 (12.29)	0.933
Age Range (%)				0.003 **
18–44	5 (3.6)	3 (4.3)	2 (2.9)	
45–64	42 (30.0)	24 (34.3)	18 (25.7)	
65–79	55 (39.3)	17 (24.3)	38 (54.3)	
80+	38 (27.1)	26 (37.1)	12 (17.1)	
Sex = Male (%)	75 (53.6)	40 (57.1)	35 (50.0)	0.498
Smoking (%)	42 (30.0)	23 (32.9)	19 (27.1)	0.58
Mental Health (%)	19 (13.6)	9 (12.9)	10 (14.3)	1.000
Oncological Diagnosis (%)	109 (77.9)	52 (74.3)	57 (81.4)	0.416
Baseline treatment (%)	114 (81.4)	54 (77.1)	60 (85.7)	0.277
>60 MME	57 (40.7)	35 (50.0)	22 (31.4)	0.039 *
0–60 MME	57 (40.7)	19 (27.1)	38 (54.3)	0.002 **
0 MME	26 (18.6)	16 (22.9)	10 (14.3)	0.277
Appropriateness to indication (%)	46 (32.9)	25 (35.7)	21 (30.0)	0.589
Concomitant use with:				
Benzodiazepines (%)	40 (28.6)	17 (24.3)	23 (32.9)	0.35
Z drugs (%)	1 (0.7)	1 (1.4)	0 (0.0)	1.000
Antirheumatic (%)	1 (0.7)	1 (1.4)	0 (0.0)	1.000
Antidepressants (%)	31 (22.1)	8 (11.4)	23 (32.9)	0.004 **
Beta-blockers (%)	10 (7.1)	7 (10.0)	3 (4.3)	0.325
Oral antidiabetics (%)	7 (5.0)	3 (4.3)	4 (5.7)	1.000
Gabepentinoids (%)	28 (20.0)	18 (25.7)	10 (14.3)	0.139
CYP3A4 inhibitors (%)	11 (7.9)	8 (11.4)	3 (4.3)	0.209
Month (%)				0.261
January	12 (8.6)	4 (5.7)	8 (11.4)	
February	15 (10.7)	5 (7.1)	10 (14.3)	
March	8 (5.7)	6 (8.6)	2 (2.9)	
April	15 (10.7)	8 (11.4)	7 (10.0)	
May	13 (9.3)	5 (7.1)	8 (11.4)	
June	4 (2.9)	3 (4.3)	1 (1.4)	
July	12 (8.6)	6 (8.6)	6 (8.6)	
August	9 (6.4)	8 (11.4)	1 (1.4)	
September	11 (7.9)	5 (7.1)	6 (8.6)	
October	14 (10.0)	6 (8.6)	8 (11.4)	
November	17 (12.1)	10 (14.3)	7 (10.0)	
December	10 (7.1)	4 (5.7)	6 (8.6)	
Hospitalization Ward (%)				0.365
No hospitalization	47 (33.6)	18 (25.7)	29 (41.4)	
General and Digestive Surgery	4 (2.9)	2 (2.9)	2 (2.9)	
Vascular Surgery	3 (2.1)	1 (1.4)	2 (2.9)	
Palliative care	21 (15.0)	13 (18.6)	8 (11.4)	
Geriatrics	6 (4.3)	2 (2.9)	4 (5.7)	
Hematology	3 (2.1)	2 (2.9)	1 (1.4)	
Internal Medicine	8 (5.7)	4 (5.7)	4 (5.7)	
Nephrology	2 (1.4)	2 (2.9)	0 (0.0)	
Neurosurgery	1 (0.7)	1 (1.4)	0 (0.0)	
Oncology	35 (25.0)	18 (25.7)	17 (24.3)	
Radiation Oncology	2 (1.4)	2 (2.9)	0 (0.0)	
Psychiatry	1 (0.7)	0 (0.0)	1 (1.4)	
Traumatology	3 (2.1)	3 (4.3)	0 (0.0)	
Urology	4 (2.9)	2 (2.9)	2 (2.9)	
Cause of pain (%)				0.551
Car Accident	1 (0.7)	1 (1.4)	0 (0.0)	
Adrenal crisis	1 (0.7)	1 (1.4)	0 (0.0)	
Diverticulitis	1 (0.7)	0 (0.0)	1 (1.4)	
Post-surgical spinal cord pain	1 (0.7)	0 (0.0)	1 (1.4)	
Pain secondary to treatment	2 (1.4)	1 (1.4)	1 (1.4)	
COPD	1 (0.7)	1 (1.4)	0 (0.0)	
Fracture	2 (1.4)	1 (1.4)	1 (1.4)	
Heart failure	3 (2.1)	2 (2.9)	1 (1.4)	
Lumbago	17 (12.1)	8 (11.4)	9 (12.9)	
Obstruction	1 (0.7)	1 (1.4)	0 (0.0)	
Oligoarthritis	1 (0.7)	1 (1.4)	0 (0.0)	
Oncological	99 (70.7)	47 (67.1)	52 (74.3)	
Chronic pancreatitis	3 (2.1)	3 (4.3)	0 (0.0)	
Colon perforation	1 (0.7)	0 (0.0)	1 (1.4)	
PET/Polyneuropathy	1 (0.7)	1 (1.4)	0 (0.0)	
Uropathy	1 (0.7)	1 (1.4)	0 (0.0)	
Vascular	4 (2.9)	1 (1.4)	3 (4.3)	

DDD, defined daily doses. ER, emergency room. MME, morphine milligram equivalents. TIRF, transmucosal immediate-release fentanyl. NSAID, non-steroidal anti-inflammatory Drug. Z drugs refers to a group of nonbenzodiazepine hypnotic medications. *, ** and *** represent <0.05, <0.01, and <0.001 significance, respectively.

**Table 2 pharmaceuticals-17-01609-t002:** Appropriateness analyzed in subpopulations before emergency room visit and at discharge.

Category	Appropriateness Prior (%)	Appropriateness at Discharge (%)
All	32.86	42.50
2019	35.71	42.11
2022	30.00	42.86
Prior TIRF		
Yes	48.78	55.07
No	10.34	25.49
Hospitalization		
No	23.40	27.27
Yes	37.63	51.32
TIRF Changes (%)		
Increase	20.00	40.00
Deprescription	36.36	63.64
Deaths	45.00	-
Prescription	5.00	25.00
Reduction	57.14	57.14
No changes	36.11	43.06
Age range (%)		
Age 18–44	40.00	50.00
Age 45–64	28.57	39.47
Age 65–79	36.36	48.98
Age 80+	31.58	34.48

TIRF, transmucosal immediate-release fentanyl.

## Data Availability

The original data presented in the study are openly available in FigShare at (https://doi.org/10.6084/m9.figshare.27635745 (accessed on 8 November 2024)).

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
