# Peer review of "Appropriateness of Prescribing Transmucosal Immediate-Release Fentanyl in the Emergency Room, During Hospitalization, and at Discharge: A Retrospective Study"

_pharmaceuticals, 2024, doi:10.3390/ph17121609_

Round 1

Reviewer 1 Report

Comments and Suggestions for Authors

The review paper by Punjabi and Ramirez, evaluates the suitability of transmucosal immediate-release fentanyl (TIRF) prescriptions in a Madrid emergency room from 2019 to 2022, following a 2018 warning about off-label use. The study underscores the prevalence of inappropriate TIRF prescriptions in the emergency room, highlighting low pre-hospital opioid dosing as a significant factor and indicating a need for improved adherence to prescribing guidelines. This review is well-written, thorough, and thematically suitable for pharmaceuticals. However, some major revisions should be addressed before publication.

Major comments:

1.    I think the reviewers should include some data from additional countries, potentially outside of Europe, to demonstrate whether this is a specific phenomenon to Spain or European countries.

Author Response

Reviewer 1

The review paper by Punjabi and Ramirez, evaluates the suitability of transmucosal immediate-release fentanyl (TIRF) prescriptions in a Madrid emergency room from 2019 to 2022, following a 2018 warning about off-label use. The study underscores the prevalence of inappropriate TIRF prescriptions in the emergency room, highlighting low pre-hospital opioid dosing as a significant factor and indicating a need for improved adherence to prescribing guidelines. This review is well-written, thorough, and thematically suitable for pharmaceuticals. However, some major revisions should be addressed before publication.

Major comments:

  1. I think the reviewers should include some data from additional countries, potentially outside of Europe, to demonstrate whether this is a specific phenomenon to Spain or European countries.

Response:

Esteemed reviewer 1,

Thank you for your thorough evaluation of our manuscript and your valuable feedback. We appreciate the opportunity to address your concerns regarding the global context and comparison to other countries. We are grateful for your insights, which have helped us improve the clarity and impact of our manuscript.

We have incorporated the following changes to the manuscript based on your suggestions:

  • Introduction: Added specific data highlighting the role of Europe and the USA in legal fentanyl consumption to provide a broader context for our study (pg. 2, lines 48-51).
  • Introduction: Included further information from previous studies in France and the Netherlands to strengthen the international comparison and highlight the scope of the issue (pg. 2, lines 52-56).
  • Discussion: Expanded the discussion to include a comparison with the USA TIRF REMS program, emphasizing the stricter requirements for documenting opioid tolerance and the potential implications for patient safety (pg. 12, lines 277-281).

We believe these additions provide a more comprehensive understanding of the global context of TIRF prescribing practices and strengthen the implications of our findings.

Regards,

Dr. G Punjabi

Dr. E Ramírez

Reviewer 2 Report

Comments and Suggestions for Authors

This study presented an interesting retrospective study investigating the prescribing patterns and appropriateness of transmucosal immediate-release fentanyl (TIRF) in the emergency room, during hospitalization, and at discharge. The authors have tackled an important and timely issue, given the risks associated with opioid misuse and the need for precise clinical indications, particularly for potent opioids like TIRF. Several minor issues relate to manuscript structure and contents are required for revision prior to publication.

1. A more explicit link between the observed prescribing patterns and potential clinical outcomes would enhance the impact of this study. Specifically, elaborating on the clinical implications of inappropriate TIRF prescribing for patient safety and long-term opioid dependence could strengthen the introduction and discussion.

2. The results section offers a thorough statistical analysis. However, discussing the clinical relevance of statistically significant findings would be beneficial. For instance, while some changes in prescribing patterns between 2019 and 2022 were statistically significant, the clinical implications—such as the effect of low-dose opioid therapy at discharge on patient pain management and quality of life—could be discussed in greater depth.

3. The authors have thoroughly discussed the limitations. It would be more insightful to discuss how these limitations might affect the generalizability of the findings and suggest more specific future research avenues, such as prospective studies or multi-center trials

4. The inclusion of additional visual elements, like comparison charts or tables summarizing key findings, would improve the readability and accessibility of the results. For instance, a summary table highlighting the main differences in TIRF prescribing appropriateness across settings and over time could be useful.

5. Data Availability Statement, Acknowledgement, and Conflicts of Interest, should be revised.

Author Response

er 2,

This study presented an interesting retrospective study investigating the prescribing patterns and appropriateness of transmucosal immediate-release fentanyl (TIRF) in the emergency room, during hospitalization, and at discharge. The authors have tackled an important and timely issue, given the risks associated with opioid misuse and the need for precise clinical indications, particularly for potent opioids like TIRF. Several minor issues relate to manuscript structure and contents are required for revision prior to publication.

  1. A more explicit link between the observed prescribing patterns and potential clinical outcomes would enhance the impact of this study. Specifically, elaborating on the clinical implications of inappropriate TIRF prescribing for patient safety and long-termo pioid dependence could strengthen the introduction and discussion.

Response:

Esteemed Reviewer 2,

Thank you for your insightful feedback. We agree that a stronger link between observed prescribing patterns and potential clinical outcomes would enhance the manuscript's impact. We have addressed your concerns by further emphasizing the clinical implications of inappropriate TIRF prescribing in both the introduction and discussion.

  • Introduction: We have expanded this section to explicitly link off-label TIRF use with an increased risk of adverse events, including respiratory depression, overdose, and death, particularly in opioid-nontolerant patients. We have also highlighted the potential for long-term opioid dependence and the challenges associated with managing withdrawal symptoms (pg. 2, lines 61-67).
  • Discussion: We have strengthened the discussion by drawing a clearer connection between the high prevalence of off-label TIRF prescribing and the potential for patient harm. We have also highlighted the discrepancy between our findings and the stricter FDA TIRF Risk Evaluation and Mitigation Strategy (pg. 12, lines 271-284).

  1. The results section offers a thorough statistical analysis. However, discussing the clinical relevance of statistically significant findings would be beneficial. For instance, while some changes in prescribing patterns between 2019 and 2022 were statistically significant, the clinical implications—such as the effect of low-dose opioid therapy at discharge on patient pain management and quality of life—could be discussed in greater depth.

Response:

Esteemed Reviewer 2,

Thank you for highlighting the need to discuss the clinical relevance of our findings. We agree that exploring the potential implications of changes in prescribing patterns on patient pain management and quality of life would strengthen the manuscript.

Acknowledging limitations: We have explicitly acknowledged the limitations of our study in not directly measuring pain and quality of life (pg 13, line 329). Furthermore, we have emphasized the need for future research to incorporate these patient-centered outcomes using validated assessment tools to gain a more comprehensive understanding of the clinical impact of TIRF prescribing practices (pg 13, lines 334-336).

Expanding the discussion: We have elaborated on the potential impact of low-dose opioid therapy at discharge on patient outcomes. This practice raises concerns about opioid tolerance, as a significant proportion of these prescriptions would be considered invalid under stricter FDA guidelines. Essentially, clinicians are prescribing TIRF to potentially opioid-nontolerant patients, putting them at risk of serious complications. This may also lead to inadequate pain control and the need for closer monitoring.  Telemedicine could play a valuable role in facilitating this ongoing monitoring and support. Surveying medical professionals about their TIRF prescribing patterns could provide valuable insights into the factors driving these practices and inform interventions to improve patient safety (pg. 12, lines 267-285). Finally, patients requiring frequent TIRF rescues should be encouraged to consult their primary care physician, utilizing telemedicine when feasible, to ensure appropriate pain management and address potential concerns about opioid tolerance (pg. 14, lines 370-372).

  1. The authors have thoroughly discussed the limitations. It would be more insightful to discuss how these limitations might affect the generalizability of the findings and suggest more specific future research avenues, such as prospective studies or multi-center trials

Response:

Esteemed Reviewer 2,

Thank you for your suggestion to elaborate on future research avenues. We agree that addressing the limitations of our study with specific recommendations for future research would strengthen the manuscript.

We have revised the discussion section (pg. 13, lines 337-341) to include more specific suggestions for future research, such as:

  • Prospective, multi-center studies: To enhance the generalizability of our findings, we recommend conducting prospective studies across multiple centers with diverse patient populations. This will help to validate our results in different clinical settings and provide a more comprehensive understanding of TIRF prescribing practices.
  • Incorporation of validated assessment tools: Future studies should incorporate validated pain management and quality of life assessment tools to capture the patient experience and evaluate the clinical impact of TIRF prescribing practices on these important outcomes.
  • Qualitative studies: To gain deeper insights into the factors influencing prescribing decisions, we suggest conducting qualitative studies, such as interviews or focus groups with clinicians, to explore their perspectives, experiences, and challenges related to TIRF prescribing.

  1. The inclusion of additional visual elements, like comparison charts or tables summarizing key findings, would improve there adability and accessibility of the results. For instance, a summary table highlighting the main differences in TIRF prescribing appropriateness across settings and over time could be useful.

Response: We agree that a summary table would be beneficial and have therefore included Table 2, which presents appropriateness data across different categories before the ER visit and at discharge (Table 2).

Table 2. Appropriateness analyzed in subpopulations before emergency room visit and at discharge.

Category

Appropriateness prior (%)

Appropriateness at discharge (%)

All

32.86

42.50

    2019

35.71

42.11

    2022

30.00

42.86

Prior TIRF

     Yes

48.78

55.07

     No

10.34

25.49

Hospitalization

    No

23.40

27.27

    Yes

37.63

51.32

TIRF Changes (%)

    Increase

20.00

40.00

    Deprescription

36.36

63.64

    Deaths

45.00

-

    Prescription

5.00

25.00

    Reduction

57.14

57.14

    No changes

36.11

43.06

Age range (%)

    Age 18-44

40.00

50.00

    Age 45-64

28.57

39.47

    Age 65-79

36.36

48.98

    Age 80+

31.58

34.48

TIRF, transmucosal immediate-release fentanyl.

  1. Data Availability Statement, Acknowledgement, and Conflicts of Interest, should be revised.

Response: Data Availability Statement, Acknowledgement, and Conflicts of Interest have been revised and corrected.

Data Availability Statement: The original data presented in the study are openly available in FigShare at (dx.doi.org/10.6084/m9.figshare.27635745).

Regards,

Dr. G Punjabi

Dr. E Ramírez

Reviewer 3 Report

Comments and Suggestions for Authors

The authors in the current work evaluated the appropriateness of transmucosal immediate-release fentanyl (TIRF) prescriptions in a Madrid emergency room during 2019 and 2022, following a 2018 warning about off-label use. This study found inappropriate TIRF prescriptions were common in an emergency room setting, often due to low pre-hospital opioid doses. While hospitalization improved TIRF appropriateness in survivors, especially in 2022, concerning prescribing practices persisted. This emphasizes the need for better education and interventions to ensure safe and effective TIRF use.

It was a good literature survey of the concerning problem with satisfactory results. The manuscript can be accepted for publication in its current form after addressing the following minor concern.

1) In the future directions section, the authors should emphasize more on how the problem can be better resolved other than just through educational programs.

Author Response

Reviewer 3,

The authors in the current work evaluated the appropriateness of transmucosal immediate-release fentanyl (TIRF) prescriptions in a Madrid emergency room during 2019 and 2022, following a2018 warning about off-label use. This study found inappropriate TIRF prescriptions were common in an emergency room setting, often due to low pre-hospital opioid doses. While hospitalization improved TIRF appropriateness in survivors, especially in 2022,concerning prescribing practices persisted. This emphasizes the need for better education and interventions to ensure safe and effective TIRF use.

It was a good literature survey of the concerning problem with satisfactory results. The manuscript can be accepted for publication in its current form after addressing the following minor concern.

  1. In the future directions section, the authors should emphasize more on how the problem can be better resolved other than just through educational programs.

Response:

Esteemed Reviewer 3,

Thank you for your suggestion to expand the discussion on potential solutions beyond educational programs. We agree that exploring a broader range of interventions is important for addressing the issue of inappropriate TIRF prescribing.

In addition to educational programs, we have highlighted the following strategies in the discussion section:

  • Improved record-tracking of off-label: We emphasize the need for better monitoring and documentation of off-label TIRF prescriptions to identify trends, assess potential risks, and inform interventions. This could involve implementing electronic health record systems that flag off-label use or require justification for such prescriptions (pg. 13, lines 363-367).
  • Predetermined dispensing quantities through electronic medical records: We suggest utilizing electronic medical records to establish standardized prescribing protocols and limit the quantity of TIRF dispensed per prescription. This could help prevent overprescribing and reduce the risk of misuse or diversion (pg 14, lines 367-370).
  • Enhanced patient-physician communication and follow: We recommend encouraging patients who require frequent TIRF rescues to consult their primary care physician. This allows for closer monitoring of pain management, assessment of opioid tolerance, and timely adjustments to treatment plans. Telemedicine can facilitate these consultations, improving access to care and enhancing communication (pg. 14, lines 370-373).

Regards,

Dr. G Punjabi

Dr. E Ramírez

Round 2

Reviewer 1 Report

Comments and Suggestions for Authors

The review paper by Punjabi and Ramirez, evaluates the suitability of transmucosal immediate-release fentanyl (TIRF) prescriptions in a Madrid emergency room from 2019 to 2022, following a 2018 warning about off-label use. The study underscores the prevalence of inappropriate TIRF prescriptions in the emergency room, highlighting low pre-hospital opioid dosing as a significant factor and indicating a need for improved adherence to prescribing guidelines. This review is well-written, thorough, and thematically suitable for pharmaceuticals. The authors addressed all my concerns about the first review, I think this paper should be accepted in current form